# The Link between Activities of Hepatic 11beta-Hydroxysteroid Dehydrogenase-1 and Monoamine Oxidase-A in the Brain Following Repeated Predator Stress: Focus on Heightened Anxiety

**DOI:** 10.3390/ijms23094881

**Published:** 2022-04-28

**Authors:** Vadim Tseilikman, Maxim Lapshin, Igor Klebanov, George Chrousos, Maria Vasilieva, Anton Pashkov, Julia Fedotova, David Tseilikman, Vladislav Shatilov, Eugenia Manukhina, Olga Tseilikman, Alexey Sarapultsev, H. Fred Downey

**Affiliations:** 1School of Medical Biology, South Ural State University, 454080 Chelyabinsk, Russia; lapshin1982@yandex.ru (M.L.); klebanovii@susu.ru (I.K.); chrousos@gmail.com (G.C.); carin-shik@mail.ru (M.V.); pashkovaa@susu.ru (A.P.); julia.fedotova@mail.ru (J.F.); manukh@mail.ru (E.M.); diol2008@yandex.ru (O.T.); a.sarapultsev@gmail.com (A.S.); freddowney@yahoo.com (H.F.D.); 2School of Electronic Engineering and Computer Science, South Ural State University, 454080 Chelyabinsk, Russia; 3University Research Institute of Maternal and Child Health and Precision Medicine, National and Kapodistrian University of Athens, 11527 Athens, Greece; 4Laboratory of Neuroendocrinology, Pavlov Institute of Physiology, RAS, 199034 St. Petersburg, Russia; 5International Research Centre “Biotechnologies of the Third Millennium”, ITMO University, 191002 St. Petersburg, Russia; 6Zelman Institute of Medicine and Psychology, Novosibirsk State University, 630090 Novosibirsk, Russia; ararat.sinaev@yandex.ru; 7Basic Medicine Department, Chelyabinsk State University, 454001 Chelyabinsk, Russia; vlad.shatilov.2018@mail.ru; 8Laboratory for Regulatory Mechanisms of Stress and Adaptation, Institute of General Pathology and Pathophysiology, 125315 Moscow, Russia; 9Department of Physiology and Anatomy, University of North Texas Health Science Center, Fort Worth, TX 76107, USA; 10Institute of Immunology and Physiology, Ural Branch of the Russian Academy of Science, 620049 Ekaterinburg, Russia

**Keywords:** anxiety-like behavior, 11-βHSD-1, corticosterone, MAO-A, mathematical modeling, norepinephrine, post-traumatic stress disorder, PSS, elevated plus maze

## Abstract

We investigated the presence of a molecular pathway from hepatic 11-βHSD-1 to brain MAO-A in the dynamics of plasma corticosterone involvement in anxiety development. During 14 days following repeated exposure of rats to predator scent stress for 10 days, the following variables were measured: hepatic 11-βHSD-1 and brain MAO-A activities, brain norepinephrine, plasma corticosterone concentrations, and anxiety, as reflected by performance on an elevated plus maze. Anxiety briefly decreased and then increased after stress exposure. This behavioral response correlated inversely with plasma corticosterone and with brain MAO-A activity. A mathematical model described the dynamics of the biochemical variables and predicted the factor(s) responsible for the development and dynamics of anxiety. In the model, hepatic 11-βHSD-1 was considered a key factor in defining the dynamics of plasma corticosterone. In turn, plasma corticosterone and oxidation of brain ketodienes and conjugated trienes determined the dynamics of brain MAO-A activity, and MAO-A activity determined the dynamics of brain norepinephrine. Finally, plasma corticosterone was modeled as the determinant of anxiety. Solution of the model equations demonstrated that plasma corticosterone is mainly determined by the activity of hepatic 11-βHSD-1 and, most importantly, that corticosterone plays a critical role in the dynamics of anxiety following repeated stress.

## 1. Introduction

Psychological stress, anxiety, and post-traumatic stress disorder (PTSD) are increasingly prevalent, and they share common signs [1,2]. According to classical neuroscience, stress-induced excitation of the amygdala is a critical event in developing anxiety-related behavior. Amygdala excitation projects to the paraventricular nucleus of the hypothalamus, which activates the hypothalamic-pituitary-adrenal axis with the resulting release of catecholamines by the adrenal medulla and glucocorticoids (GCs) by the adrenal cortex, especially cortisol (in humans) or corticosterone in rodents [3]. The amygdala also communicates to the *locus coeruleus* to activate monoaminergic neurons. The resulting increase in brain monoamines, especially norepinephrine, enhances excitation, fear, and coding of the memory of fear [4]. This limbic-hypothalamic-pituitary-adrenal axis is the classical model of the physiological response to psychological stress (Figure 1).

However, in some responses to stress, such as human PTSD and rat models of PTSD, decreased plasma cortisol/corticosterone has been reported [5,6]. Recently, we showed that 10 days after rats had been exposed to predator stress, i.e., cat urine odor, for 14 days, their plasma corticosterone was negatively correlated with their level of anxiety [7]. Moreover, in this experimental model, we found that decreased corticosterone was associated with decreased brain monoamine oxidase-A (MAO-A) activity [8]. Reduced MAO-A activity leads to an increase in brain norepinephrine [8], and as stated above, sustained activation of the noradrenergic system is involved in the development of stress-induced disorders. These include anxiety disorders and PTSD, as has been demonstrated by clinical and experimental studies [8].

Stress-induced GCs boost MAO-A activity [9,10], and this ultimately decreases brain monoamines. Reduced GCs also facilitate the effects of pro-inflammatory cytokines that provoke inflammatory complications [11,12,13]. Yehuda and Seckl [3] provided evidence that during some stress-related disorders, plasma GCs are reduced due to hepatic metabolism. GCs are metabolized in the liver by 11β-hydroxysteroid dehydrogenase 1 (11-βHSD-1) and by isoforms of cytochrome P450 of the CYP3A subfamily [14]. 11-βHSD-1 induces irreversible inactivation of GCs, whereas CYP3A causes reversible inactivation.

Based on the liver’s ability to regulate plasma GC [14], recent reports have suggested that the liver plays a special role in the pathogenesis of some anxiety-related disorders. Specifically, a strong positive correlation was found between hepatic 11-βHSD-1 and anxiety behavior in a model of repeated predator stress [15].

However, the pathway from hepatic GC-metabolizing enzymes to time-dependent changes in anxiety has not been verified, nor have the kinetics of this response been elucidated. This pathway explains the paradoxical combination of low GCs with increased anxiety following some types of stress. Understanding this pathway may lead to new molecular targets for treating some stress-induced disorders, including PTSD.

Figure 1 illustrates how activation of hepatic 11-βHSD-1 would lead to reduced GCs and then to decreased brain MAO-A and increased anxiety associated with elevated brain norepinephrine. The aim of this study was to verify the presence and demonstrate the function of a liver–brain axis in the development of anxiety following repeated or chronic stress. To accomplish this aim, time-dependent, anxiety-like behavior was induced in a rat model of repeated predator stress, and correlations between an index of anxiety and concentrations/activities of hepatic enzymes of GC metabolism (11-βHSD-1 and CYP3A4), brain MAO-A, blood and brain corticosterone, and brain norepinephrine were analyzed.

Based on the experimental results, a mathematical model was developed to describe the kinetics of the relevant enzymatic reactions in the liver–brain axis. The model is based on a system of ordinary differential equations and a mathematical approach traditionally used to describe the rates of chemical processes. Thus, the model integrates processes of the liver–brain axis that modulate anxiety severity: (a) plasma corticosterone; (b) hepatic 11-βHSD-1 activity; (c) brain MAO-A activity; (d) brain norepinephrine concentration; (e) oxidation indices of brain ketodienes and conjugated trienes. The mathematical model accurately reproduced the experimentally determined values, but more importantly, it provided new information on the time-varying values of the critical reactants throughout the post-stress period.

Thus, this modeling approach allowed the identification of an expanded range of interrelations in the process of anxiety pathogenesis, and it may also indicate potential pharmacological targets for the treatment of stress-induced anxiety.

## 2. Results

The order of presentation is based on the link between glucocorticoids metabolism in the liver with heightened anxiety development in the repeated predator stress paradigm.

### 2.1. Time-Dependent Development of Anxiety-like Behavior

The behavior of rats that had experienced the stress of exposure to cat urine was evaluated in an elevated plus maze (EPM) test. These tests were performed on different groups of rats on the 3rd, 7th, 10th, and 14th days after exposure to predator stress. The behavior of different groups of unstressed control rats was evaluated after corresponding periods of cage rest with no exposure to predator stress. The results are shown in Figure 2.

The time spent by stressed rats in the open arms of the EPM decreased by 65.1% (*p* = 0.027) from the 3rd to the 7th day post stress (Figure 2A), and this time remained low at 10 and 14 days post stress. On the 10th and 14th days, the values for the stressed rats were significantly less than those of control rats, 57.9% (*p* = 0.0022) and 65.3% (*p* = 0.00018), respectively. Moreover, the value on day 14 was 80% (*p* = 1.08 × 10^−5^) less than on day 3. These data demonstrate greater anxiety-like behavior in the stressed rats, and this behavior increased during the post-stress period. As expected, the spent in closed arms by the stressed rats (Figure 2B) was inversely related to the time spent in the open arms.

The number of entries into open arms of the EPM (Figure 2C) by stressed rats decreased by 56.9% (*p* = 0.029) from the 3rd to the 7th day post stress, and these values remained low for the remainder of the protocol. On day 14, entries of stressed rats into open arms was 295% (*p* = 0.0043) less than control and was 64% (*p* = 0.0022) less than in 3rd day post stress. The number of entries into the closed arms of the EPM (Figure 2D) was higher in the control group on the 3rd day in comparison with the PS group (*p* = 0.006). This quantity increased from day 3 to day 14 for the stressed rats by 124% (*p* = 0.057) and by 41.5% (*p* = 0.28) for the control rats, but the latter results did not reach the level of statistical significance set at 0.05. On day 14, this value was 29.3% (*p* = 0.7) less for stressed rats than for control rats and was 223% (*p* = 0.0033) greater for the 3rd day post stress, respectively. The fewer entries of stressed rats into the closed arms is consistent with the longer time these rats spent in the closed arms.

The anxiety index (AI, Figure 2E, see Methods) was 27.1% (*p* = 0.0015) less for stressed rats at 3 days post stress than for control rats. This index for stressed rats then increased by 36.3% (*p* = 0.0039) from day 3 to day 7, and remained unchanged (*p* = 0.77) from day 7 to day 10. It remained high at 14 days post stress. At 10 and 14 days post stress, the AI of stressed rats was 13.6% (*p* = 0.023) and 21.4% (*p* = 1.08 × 10^−5^) greater than control, and 72% (*p* = 1.2 × 10^−5^) greater than at 3 days post stress, respectively.

### 2.2. The Dynamics of Circulating Corticosterone Concentrations, Hepatic 11-βHSD-1, and CYP3A Activities, and Protein Content

Figure 3 shows the concentration of plasma corticosterone, concentrations, and activities of 11-βHSD-1 and CYP3A, and the content of liver protein content on the 3rd, 7th, 10th, and 14th days of the post-traumatic period.

Plasma corticosterone concentration was decreased by 67.9% (*p* = 0.0002) compared to control on the 14th day of the post-traumatic period and by 66.0% (*p* = 0.008) compared to its value on the 3rd day (Figure 3A). The concentration of hepatic 11-βHSD-1 protein (Figure 3B) was decreased by 56.7% (*p* = 2.17 × 10^−5^) compared to control on the 3rd day. Contrarily, by the 14th day, this concentration had increased by 331% (*p* = 0.002), and it was then 31.6% greater than control (*p* = 0.015) and was 430% (*p* = 2.6 × 10^−5^) greater than on day 3. Simultaneously, its enzymatic activity (Figure 3C) decreased by 57.6% (*p* = 0.002) by the 10th day. By the 14th day, this activity had then increased by 92.9% (*p* = 0.04) compared to its value on day 10, so that by the 14th day, hepatic 11-βHSD-1 activity was 100% (*p* = 0.005) greater than control. The CYP3A protein concentration (Figure 3D) tended to increase from day 3 to day 10 and then decreased by 47% (*p* = 0.01), so that on day 14, its concentration was 48.1% (*p* = 0.0002) less than control. CYP3A activity (Figure 3E) followed a similar pattern so that by the 14th day, it was 58% less than the control (*p* = 0.037). Thus, the apparent suppression of the CYP3A pathway of glucocorticoid metabolism suggests that its contribution to the regulation of blood glucocorticoids in chronic PTSD could be neglected. Therefore, for the development of the mathematical model, we used only data concerning the 11βHSD pathway.

### 2.3. The Dynamics of Norepinephrine Concentration, MAO-A Activity, Protein Concentration, and Lipid Perosidase (LPO) Values in the Brain

Figure 4 shows brain norepinephrine concentration, MAO-A protein concentration, and activity, basal LPO values, and Fe^2+^/ascorbate induced LPO values on the 3rd, 7th, 10th, and 14th days of the post-traumatic period and at corresponding times in unstressed, control rats.

Brain norepinephrine (Figure 4A) was similar to control on day 3, but was 52.6% (*p* = 0.00018) less than control on day 7. Norepinephrine then increased by 453% (*p* = 0.006) to be 232% (*p* = 0.0002) greater than control on day 10. Norepinephrine then decreased by 66.5% (*p* = 0.002) to be similar to control on day 14.

The brain concentration of MAO-A protein (Figure 4B) and MAO-A activity (Figure 4C) were less than control on day 3, 23.63% (*p* = 0.029) and 34.3% (*p* = 0.01), respectively. On day 7, MAO-A increased by 69.8% (*p* = 0.002) to be 17.3% (*p* = 0.004) greater than control. On day 7, MAO-A activity increased by 72.4% (*p* = 0.002). From day 7 to day 10, MAO-A values further increased and at that time were 67.1% (*p* = 0.015) greater than control. MAO-A and MAO-A activity then decreased by 63.9% (*p* = 0.002) and 55.8% (*p* = 0.049), respectively, so that on day 14 they were 52.6% (*p* = 1.08 × 10^−5^) and 48.6% (*p* = 7.6 × 10^−5^) less than control, respectively.

Basal LPO (Figure 4D) in the brain of stressed rats was 20% (*p* = 4.33 × 10^−5^) higher than the control on day 3. On days 7 and 10, basal LPO of stressed rats was not significantly different from control. From day 10 to day 14, it increased 52.9% (*p* = 0.002), and it was 30% (*p* = 0.003) greater than the control on day 14 as well as it was 31% (*p* = 2.5 × 10^−5^) greater than on day 3.

Fe^2+^/ascorbate induced LPO (Figure 4E) of stressed rats did not differ from that of control rats at any time during the post-traumatic period. For both groups, this concentration in stressed rats increased from day 3 to day 14 by 35% (*p* = 0.019).

### 2.4. The Mathematic Model and Its Experimental Validation

We propose the following system of ordinary differential equations as a mathematical model of the relationship between relevant biochemical processes and anxiety during the post-stress syndrome.
(1)$$\frac{dc}{dt}=-\frac{a_{0}*c*b\left(t\right)}{k_{1}+c}$$
(2)$$\frac{dm}{dt}=a_{1}*c^{q_{1}}-a_{2}*i_{5}{\left(t\right)}^{q_{2}}$$
(3)$$\frac{dn}{dt}=\frac{a\left(t\right)*n*\left(m-m_{0}\right)}{k_{2}+n}$$
(4)$$\frac{dAI\;}{dt}=a_{3}*c{\left(t\right)}^{q_{3}}$$
where *a*(*t*) is the time-dependent phenomenological function,

*b*(*t*) is the enzymatic activity of hepatic 11-βHSD-1;

*c*(*t*) is the plasma concentration of corticosterone;

*i*_5_(*t*) is the oxidation indices of brain ketodienes and conjugated trienes;

*m*(*t*) is the specific activity of brain MAO-A;

*n*(*t*) is the concentration of brain norepinephrine; *AI*(*t*) is the anxiety index.

The initial values of the dependent variables were equated to the experimental values measured on the third day after stress exposure. The constants *a_i_*, *k_i_*, *q_i_*, are phenomenological coefficients.

Equation (1) in the system characterizes the circulating corticosterone concentrations during the post-stress period. Hepatic 11-βHSD-1 was considered a key factor in defining the dynamics of circulating corticosterone. As shown in Figure 3A and Figure 5A, the plasma concentration of corticosterone decreased monotonically, having reached a minimum on the 14th day after the end of the predator stress. As can be seen in Figure 5B, the calculated curve of the dynamics of circulating corticosterone concentrations almost completely coincides with the curve constructed from the experimental data. Therefore, we can assume the proposed mathematical model adequately characterizes the decrease in corticosterone concentrations during the post-stress period, at a time when anxiety increased.

Equation (2) in the system characterizes post-stress MAO-A activity. Circulating corticosterone concentrations and the oxidative degradation of lipids in the brain were considered factors determining MAO-A activity dynamics (Figure 4C and Figure 5C). Despite the value changes in the concentration of brain lipid oxidation products, its use as an additional variable made it possible to achieve agreement between the model-calculated values of MAO-A activity and the experimental values (Figure 5C,D). The solution of proposed Equation (2) well reproduced the nonlinear nature of the dynamics of MAO-A activity observed during the post-stress period. However, the calculated maximum of enzymatic activity shifted to the 12th day, whereas in the experiment, it was observed on the 10th day.

The third equation in the system characterizes the norepinephrine concentration in the brain. MAO-A activity was considered the sole factor in defining the dynamics of brain norepinephrine concentration. Again, model-calculated values (Figure 6B) closely approximated the experimentally measured values (Figure 6A).

The mathematical model reproduced the typical post-stress transition from an initial decrease to a subsequent increase in brain norepinephrine (Figure 6B). However, the calculated curve, in comparison to the experimental curve (Figure 6A), displayed a rightward shift. The calculated minimum of norepinephrine was predicted to be between the 7th and 10th days, while in the experiment, it was observed on the 10th day. The calculated maximum of norepinephrine was predicted by the model to be on the 14th day, while in the experiment, the maximum increase was detected on the 10th, followed by a subsequent return to the normal values. According to the calculated data, the maximum concentration of norepinephrine falls on the 14th day. Meanwhile, in the experiment, the maximum increase was observed only on the 10th day, followed by the normalization of the concentration of norepinephrine.

The data presented in Figure 7 show a suitable agreement between the experimental (Figure 7A) and the model-calculated curves (Figure 7B). This agreement validates the important model assumption of the dependence of AI on the plasma corticosterone concentration. Since plasma corticosterone decreased as AI increased (Figure 2E and Figure 3A), other factors were responsible, specifically, the increase in hepatic 11-βHSD-1.

## 3. Discussion

This study demonstrated the time-dependent nature of anxiety-like behavior in rats previously exposed to repeated predator-sent stress. Anxiety decreased briefly and then increased following stress exposure, with the most pronounced increase observed from the 10th to the 14th day of the post-stress period. Importantly, the brief anxiolytic reaction, evident only on the 3rd day, corresponded to measured and modeled transient decreases in brain norepinephrine.

Earlier, we found that an initial, brief, anxiolytic response following repeated stress was associated with an increase in brain GABA [7]. This anxiolytic response was also associated with initially decreased liver 11βHSD. Later, hepatic 11-βHSD-1 activity and 11-βHSD-1 protein concentration increased. A simultaneous decrease in hepatic CYP3A activity and protein concentration indicated that this pathway of glucocorticoid metabolism is suppressed in the liver of previously stressed rats. Therefore, following predator stress, 11-βHSD-1-dependent liver metabolism of glucocorticoids is predominant.

Using the hexobarbital sleep test (HST), we identified a rat phenotype that was especially sensitive to predator scent stress [16]. In these rats, the hepatic 11-βHSD-1 pathway was activated and associated with increased anxiety. In contrast, the CYP3A-dependent pathway was correlated with reduced anxiety.

Since correlation cannot confirm the existence of a causal relationship between changes in the activities of hepatic enzymes and time-varying changes in plasma corticosterone and anxiety, we used mathematical modeling to further evaluate this relationship. In that model, plasma corticosterone was considered to be the sole variable determining changes in anxiety (Equation (4)). The coincidence of the experimental values of plasma corticosterone and of an index of anxiety with those calculated from the model when using experimentally measured values of hepatic 11-βHSD-1 activities confirmed that hepatic 11-βHSD-1 is the key enzyme that breaks down plasma corticosterone during the development of anxiety. Furthermore, this conclusion agrees with the previously reported involvement of a hepatic 11-βHSD-1-dependent pathway of GC metabolism [3] and the resulting negative correlation between anxiety and plasma corticosterone during experimental predator stress [14].

The behavioral and biochemical changes that occurred from the 7th to the 10th day of the post-traumatic period merit attention. On the 7th day, brain norepinephrine was diminishing simultaneously with an increase in brain MAO-A protein concentration. Thus, it appears that at this time, low brain norepinephrine resulted in its augmented metabolism. By the 10th day, an increase in the brain norepinephrine was observed, and heightened anxiety was also preserved on the 14th day. This agrees with our previous finding of increased norepinephrine in the cortex, hippocampus, medulla oblongata, and cerebellum during stress-induced anxiety [16].

MAO-A is involved in the regulation of brain norepinephrine [17]. In our model, time-dependent changes in brain norepinephrine are solely determined by MAO-A activity (Equation (3)). Again, experimental and calculated values describing time-dependent changes in brain norepinephrine are in suitable agreement. This further supports the view that MAO-A is an important regulator of brain norepinephrine in rats following predator scent stress.

In turn, the activity and expression of MAO-A are regulated by glucocorticoids [12]. We hypothesized and modeled that plasma corticosterone mediates the relationship between hepatic 11-βHSD-1 and brain MAO-A activities. This hypothesis was based on observations of GC-dependent regulation of MAO-A activity under chronic stress conditions [18,19]. We also considered that GC can have a bidirectional effect on MAO-A activity [20]. The ability of GC to affect the expression of the MAO-A gene is well known. However, the activity of MAO-A, as a mitochondrial outer membrane-bound enzyme, can be reduced due to the direct action of GC on the membrane [21]. MAO-A activity depends on many other membrane processes, including lipid peroxidation [22], the intensity of which is also altered by GC [23]. It is noteworthy that by combining the concentration of corticosterone concentration with the content of LPO products in the equation characterizing MAO-A activity Equation (2), it was possible to accurately characterize the dynamics of MAO-A activity in the stressed rats of the current study. The similarity of model values with the experimental data confirmed this assumption.

In the differential equation Equation (2) describing time-dependent changes in brain MAO-A activity, plasma corticosterone concentration was represented as one of two variables. The second variable was the amount of lipid peroxidation products. MAO-A is a mitochondria membrane enzyme, and its activity is sensitive to the lipid microenvironment [24]. The model reflected well this particularity of MAO-A activity.

The negative correlation between LPO products and MAO-A activity in the brain [23] is due to a change in the conformation of the enzyme and its partial cleavage under the action of free radicals [24]. However, a model that took into account only the effect of lipid peroxidation could not accurately reproduce the experimental dynamics of MAO-A activity. Only by considering the joint effect of corticosterone and lipid peroxidation was it possible to obtain MAO-A activity values close to the experimental data [4,8].

Although other studies focused on the stress-related fluctuations of blood GC [25] or brain norepinephrine [26], this is the first study to examine the association and interaction of these variables. Our model demonstrates the dependency between reduced GC and increased brain norepinephrine in response to repeated stress. We have demonstrated that reduced blood GC can cause low MAO-A enzyme activity in the brain and, consequently, an increase in brain norepinephrine. Finally, our model demonstrates a link between liver and brain enzymes (Figure 8) in the pathogenesis of anxiety-like behavior following repeated stress.

Future studies should examine hepatic function in other experimental models of repeated stress, such as the stress-restress paradigm. This would ensure that the findings presented here are not particular features of one animal model but that they are universal. However, indirect evidence in favor of the universality of this pattern was obtained earlier in an immobilization stress model [27]. Further studies should also focus on other neuronal factors sensitive to the liver–brain axis. In this regard, it would be important to consider the link between norepinephrine and neurotrophins in repeated predator stress. The relevance of this approach is illustrated by a strong negative correlation between the amount of brain-derived neurotrophic factor (BDNF) in the hippocampus and the extent of anxiety-related behavior in the light/dark preference test, as reported by Yamamori and coauthors [28]. In turn, the excitatory effect of glutamate was also accompanied by a decrease in BDNF [29]. In addition, norepinephrine potentiation of glutamate excitatory transmitter action has been reported [30].

To further confirm a link between glucocorticoid metabolism objectively in the liver and heightened anxiety, a highly applicable mathematical model was developed. Although this model demonstrates the link between liver and brain enzymes, this is still not absolute proof that the liver–brain enzyme duet is critically involved in the development of heightened anxiety. However, the close agreement between the experimental and calculated curves augments the probability that the liver–brain axis is present. New experimental studies are required to verify this molecular pathway. Therefore, the current mathematical model may be only a first step in this direction. The next step should include experimental modulation by selective inhibitors of hepatic 11-βHSD-1 activity in stressed rats and examination of the effects on neuro-endocrine and behavior factors. In addition to the effects on brain norepinephrine concentrations, it will be important to evaluate the effects on relevant metabolites. Recently, the combinations of gas chromatography-quadrupole time of flight mass spectrometry (GC-Q-TOF/MS) and liquid chromatography-quadrupole time of flight mass spectrometry (LC-Q-TOF/MS) have been applied successfully in mechanistic metabolic studies to achieve more sensitive and accurate metabolite profiles [31]. Thus, GC-Q-TOF/MS and LC-Q-TOF/MS may be valuable tools in future studies to validate the liver–brain axis.

Our results should be expanded in further studies of the pathogenesis of anxiety disorders. Specifically, hypocorticoidism facilitates neuroinflammation [5], and, in turn, neuroinflammation is involved in the pathogenesis not only of anxiety disorders but also of PTSD, depression, and schizophrenia [5,15]. Thus, it would be important to investigate the contribution of neuroinflammation to dysfunction of the prefrontal cortex, which develops in anxiety and depressive disorders of variable etiology [2,32,33,34,35,36,37,38,39]. Since pharmacotherapy of anxiety disorders is complex, other therapeutic options, such as non-invasive brain stimulation (NIBS), seem promising [40,41]. As new therapies are evaluated, their effects on the severity of neuroinflammation should be considered.

## 4. Materials and Methods

### 4.1. Animals and Ethical Permissions

A total of 80 adult male Wistar rats were used in these experiments. The rats were housed in individually ventilated, standard cages (3–4 rats/cage) and received a high-quality rat chow (Beaphar Care Plus Rat Food) and tap water ad libitum. The temperature (22–25 °C) and humidity (55%) of the vivarium were controlled, and a 12:12 h light–dark cycle was maintained with lights on between 7:00 and 19:00. The same person handled the rats for 14 days before the start of experimental procedures and for the duration of the experimental protocols. This person also performed the predator stress and sham stress procedures and the behavioral testing. All animal procedures were performed in accordance with the U.S. National Research Council Guide for the Care and Use of Laboratory Animals (publication 85-23, revised 2011), and the experimental protocols were approved by the Animal Care and Use Committee of the Institute of General Pathology and Pathophysiology, Moscow, Russia (Project 0520-2019-0030) and by the Ethical Committee for Animal Experiments of South Ural State University, Chelyabinsk, Russia (project 0425-2018-0011, 17 May 2018, protocol number 27/521).

### 4.2. Modeling of Stress-Induced Anxiety

A paradigm of repeated exposure to predator scent stress (PSS) was performed as previously described in earlier investigations of experiment PTSD [14,15,16]. Rats were randomly assigned to eight groups (four groups of experimental rats and four groups of control rats) of 10 each. After 14 days of cage rest, the following protocols were followed. (1) For 10 days, all experimental rats were exposed daily for 10 min to PSS [15]. For 10 days, all control rats were exposed daily for 10 min to sham PSS (tap water). (2) At 3, 7, 10, and 14 days after PSS or sham PSS, experimental groups 1–4, respectively, and control groups 1–4, respectively, were tested for behavior and then killed for biochemical analyses of blood, brain and liver variables. Figure 9 illustrates the experimental timeline.

### 4.3. Behavioral Assessment

On the 15th day after the last exposure to PSS, the anxiety level of the rats was measured using the elevated plus maze (EPM) test, as previously described [42]. The test lasted ten minutes. The AI was calculated using the formula:(5)AI=1−TopT +NopN 2

*T_op_*—time in open arms

*T*—total time on maze

*N_op_*—number of entries into open arms

*N*—number of all entries

### 4.4. Evaluation of Plasma Corticosterone Concentration

Plasma CORT concentrations were determined using an enzyme-linked immunosorbent assay (ELISA) kit for measurement of CORT (Cusabio ELISA Kit, Houston, TX, USA) as per the manufacturer’s instructions. The assay sensitivity was 0.25 ng/mL, and the intra- and inter-assay coefficients of variation were both <5%.

### 4.5. Evaluation of Hepatic 11-βHSD-1 and CYP3A Protein Concentration and Enzymatic Activities

Hepatic 11-βHSD-1 and CYP3A protein concentrations were measured using a rat ELISA kit (Blue Gene Biotech, Shanghai, China) according to the manufacturer’s instructions. The assay sensitivity was 1.0 ng/mL, and the intra- and inter-assay coefficients of variation were both <5%. The hepatic 11-βHSD-1 activity was evaluated by a decrease in 10 μM corticosterone. A total of 0.1 M sodium phosphate buffer (pH 8.5) with 1.5 mM NADP was used. Incubation of the samples was conducted for 60 min at 37 °C. The sample containing the substrate (corticosterone) was added after the end of incubation, and the blank sample containing an equivalent volume of solvent was incubated simultaneously [27]. Changes in fluorescent intensity (405 nm excitation and 546 nm emission wavelengths) were measured using a VERSA FLUOR spectrofluorometer (Bio-Rad, Hercules, CA, USA).

The hepatic CYP3A activities were evaluated as described previously [27]. Briefly, livers were homogenized in 1.15% KCl. The homogenates were centrifuged at 9000× *g* for 20 min, followed by 60 min centrifugation of the supernatant at 100,000× *g*. Microsomal pellets were resuspended in 0.1 M Tris-HCl buffer (pH 7.4) containing 0.5 mM dithiothreitol, 0.1 mM EDTA, and 20% glycerol. Microsomal protein concentrations were determined by the Bradford protein assay method, using the Bio-Rad Protein Assay kit (Bio-Rad, Hercules, CA, USA) and bovine serum albumin (BSA; Sigma-Aldrich Inc., St. Louis, MO, USA) as the standard, according to the protocol provided by the manufacturer. The total activity of CYP3A was determined by measuring the amount of formaldehyde formed in the reaction of CYP3A-dependent N-demethylation of erythromycin [43]. The reaction system contained 50 mM potassium phosphate buffer (pH¼ 7.4), 3 mM MgCl_2_ (Fluka, Buches, Switzerland), 12.5 mM/L erythromycin (Sigma-Aldrich, St-Louis, MO, USA), and 0.5–1 mg microsomal protein. The reaction was started using 0.25 mM NADP (Merck, Darmstadt, Germany) and ended with the samples being placed on ice. The samples were then centrifuged after adding 200 L of 15% trichloracetic acid. Formaldehyde concentration was measured in the supernatant spectrophotometrically (405 nm) using the Nash’s reagent containing 2 M ammonium acetate, 0.05 M glacial acetic acid, and 0.02 M acetylacetone.

### 4.6. Evaluation of MAO-A Activity

Brain MAO-A activity was evaluated in tissue homogenates according to Tipton et al. [44]. Brain tissue homogenates were preincubated with 100 µL of 0.5 µM L-deprenyl, a selective inhibitor of MAO-B, for 60 min at 37 °C, and then a specific MAO-A substrate, 5-hydroxytryptamine creatinine sulfate (4 mM), was added. For inhibition of MAO-B activity, 100 µL of 1 µM clorgyline was added to 1 mL of mitochondrial suspension containing MAO in the mem-brane-bound form and incubated for 60 min at 37 °C.

Isolation of mitochondria from brain tissue homogenate was conducted according to Satav and Katyare [45]. To determine MAO-A activity under conditions of induction of free radical oxidation, mitochondrial suspensions were then preincubated for 15 min at 370 °C with 1 µM Fe^2+^ and 0.5 mM ascorbate. MAO activity was measured spectrophotometrically and expressed as nM serotonin/mg protein/min.

### 4.7. Evaluation of Norepinephrine Concentrations

Norepinephrine concentrations were measured in the whole brain. Brain tissues were homogenized in 0.1 M perchloric acid. After homogenization, the samples were centrifuged (7000× *g* for 15 min at 4 °C), and the supernatants were filtered through a syringe filter (0.2-micron pore size; Whatman, Marlborough, MA, USA) before HPLC analysis on a Shimadzu LC-20 Prominence Chromatographic System (Shimadzu, Kyoto, Japan). HPLC analysis was performed on a C18 reversed-phase column BDS Hypersil (250 × 4.6 mm, particle size. 5 µm) under isocratic conditions, with electrochemical detection. The mobile phase consisted of a 75 mM phosphate buffer containing 2 mM citrate acid, 0.1 mM octane-1-sulfonic acid, and 15% (*v*/*v*) acetonitrile (pH 4.6). Electrochemical detection (DECADE II, Antec Scientific, Zoeterwoude, The Netherlands) was achieved by setting a glassy carbon working electrode at +780 mV. The final concentration of norepinephrine was expressed as ng/µg wet tissue using an external calibration curve.

### 4.8. Evaluation of Oxidative Stress

Brain LPO content was evaluated with a spectrophotometric method [46]. This method allows differential measurement of acyl peroxides among phospholipids extracted from propanol-2. To evaluate the intensity of the induced lipid peroxidation, a mixture of 0.5 mM ascorbic acid with 50 μg FeSO4 was added to the propanol-2 extracts. Then, after 10 min, when the greatest change in the content of molecular products had been observed, spectrophotometric determination of diene conjugates, as well as ketodienes and conjugated trienes, was performed. Results were expressed as oxidation indices: E232/220 for relative contents of conjugated dienes, E278/220 for ketodienes, and conjugated trienes. The content of these lipid peroxidation products was measured before and after the addition of the Fe^2+^/ascorbate mixture, which is considered the lipid peroxidation inductor, into the extracts.

#### 4.8.1. Mathematical Modeling

Our mathematic model is based on the following facts:

(1) The ability of 11-βHSD-1, a bidirectional enzyme with predominant oxoreductase activity, to regulate GC concentrations was established as an important mechanism in the pathogenesis of PTSD [12,47,48].

(2) MAO-A enzyme, as a protein of the outer mitochondrial membrane, is sensitive to changes in its phospholipid microenvironment and the process of lipid peroxidation [48,49]. In stressful situations, GC-dependent modulation of MAO-A activity is extremely important. MAO-A expression is known to be increased by GCs. However, by increasing lipid peroxidation processes, GCs can also reduce their activity [23,24].

(3) Being one of the crucial neurochemical messengers in the CNS, norepinephrine is synthesized in the *locus coeruleus* (LC) of the brainstem, from where it is released by axonal varicosities throughout the brain [50]. Released norepinephrine activates receptors located on the postsynaptic membranes and causes postsynaptic reactions. After reuptake by the norepinephrine transporter in the synaptic cleft, most norepinephrine is metabolized in the mitochondria by MAO [32].

(4) The interplay of longitudinal changes in the arousal and sympathetic systems and the HPA axis (norepinephrine and cortisol) may underlie the natural history and pathophysiology of PTSD [33].

#### 4.8.2. Processes Being Modeled

Oxidation of the corticosterone in the nervous tissue of stressed rats by the enzyme 11-beta-HSDH-1 over the period of 12 days, i.e., the 3rd to the 14th day after a ten-day period, while rodents were regularly subjected to predator stress.

Changes in the activity of the enzyme monoamine oxidase-A in the mitochondria of the brain tissue of rats under the influence of the corticosterone and oxidative destruction of the mitochondrial membrane from the 3rd to the 14th day after the exposure to stress.

Oxidation of norepinephrine in the nervous tissue of the brain of rats by the MAO-A enzyme during the period from the 3rd to the 14th days after the exposure to stress

The influence of corticosterone on anxiety levels was assessed by behavioral patterns in rats during the EPM tests (the elevated plus maze).

##### Oxidation of the Corticosterone in the Nervous Tissue by the Enzyme 11-β-HSDH-1

Enzymatic reaction catalyzed by 11-β-HSDH-1:

corticosterone + NADP^+^ → 11-dehydrocorticosterone + H^+^ + NADPH

must be described by the Michaelis–Menten equation for non-allosteric enzymes:
(6)$$V_{0}=\frac{k_{2}*E_{0}*S}{K_{m}+S}$$

We can develop the following asymptotic model based on the equation above.
(7)$$V_{0}=-\frac{dc}{dt}\;\Rightarrow \;\frac{dc}{dt}=-\frac{a_{0}*b\left(t\right)*c}{k_{1}+c}$$
where the concentration of corticosterone, *c*, corresponds to the substrate *S*; the concentration of 11-β-HSDH-1, *b*(*t*), corresponds to the concentration of the enzyme *E*_0_, *k*_1_ corresponds to the Michaelis constant Km for the enzyme 11-β-HSDH-1; *a*_0_ corresponds to *k*_2_ the catalytic constant of the enzyme 11-β-HSDH-1.

Function *b*(*t*) was obtained from four values of enzyme concentration on the 3rd, 7th, 10th, and 14th days using the spline interpolation method on Maple software.

The following parameter values were used for the calculation: *a*_0_ = 25, *k*_1_ = 100.

Activity of the monoamine oxidase-A in mitochondria of rat brain tissue.

We consider two factors:

The effect of blood concentration of corticosterone on the initiation of transcription of the MAO enzyme in brain tissue.

Deactivation of a membrane enzyme due to oxidative destruction of membranes. The intensity of oxidative destruction is determined by the concentration of membrane lipid oxidation products (ketodienes and trienes):(8)$$\frac{dm}{dt}=a_{1}*c^{q_{1}}-a_{2}*i_{5}{\left(t\right)}^{q_{2}}$$
where *C* is the blood corticosterone concentration; *i*_5_(*t*) is a function characterizing the intensity of lipid destruction in rat brain tissue obtained from four values of concentrations of membrane lipid oxidation products on the 3rd, 7th, 10th, and 14th days using the spline interpolation method.

For the model calculation, we used the following parameter values: *a*_0_ = 25, *k*_1_ = 100, *a*_1_ = 0.25, *q*_1_ = 0.018, *a*_2_ = 0.00115, *q*_2_ = 2.15, *k*_2_ = 0.1, *m*_0_ = 1.6.

### 4.9. Statistical Analysis

Data were analyzed with SigmaPlot 12.5 and R programming language (R version 4.1.2, 2021). Quantitative data are presented as mean ± SD. Shapiro–Wilk criterion was used to check for normality distribution, and the Mann–Whitney criterion (U test) or Wilcoxon rank-sum test (depending on whether the samples were paired) were used to compare all outcome measures between the groups. *p* < 0.05 was considered statistically significant.

## 5. Conclusions

During the period following repeated traumatic stress, anxiety increased, although plasma corticosterone fells due to an increase in liver 11-β-HSDH-1. There was a resulting decrease in brain MAO-A and an increase in brain norepinephrine along with anxiety. A mathematical model reproduced the experimentally observed relations between anxiety and liver 11-β-HSDH-1, plasma corticosterone, the intensity of glucocorticoid tissue metabolism, brain MAO-A activity, and brain norepinephrine. The model provided information on the time-varying values of the relevant variables at a frequency impossible to duplicate experimentally. This was especially true with regard to the role of hepatic glucocorticoid metabolism in the regulation of plasma corticosterone. Moreover, the proposed model may serve as a basis for future theoretical and experimental studies.

## 6. Limitations

Although the predictions of our model were remarkably consistent with the experimental data, this does not exclude the possibility that a modified or different model might also produce results consistent with our or other experimental data. It is possible, or even likely, that brain mechanisms of anxiety-like behavior are not restricted to the elevation of norepinephrine. In this regard, interactions between monoamine neurotransmitters and changes in cortex serotonin should also be considered. Thus, further research may yield a more complete and more complex model of the hepatic-brain axis and its role in the behavioral response to repeated stress.

## Figures and Tables

**Figure 1 ijms-23-04881-f001:**
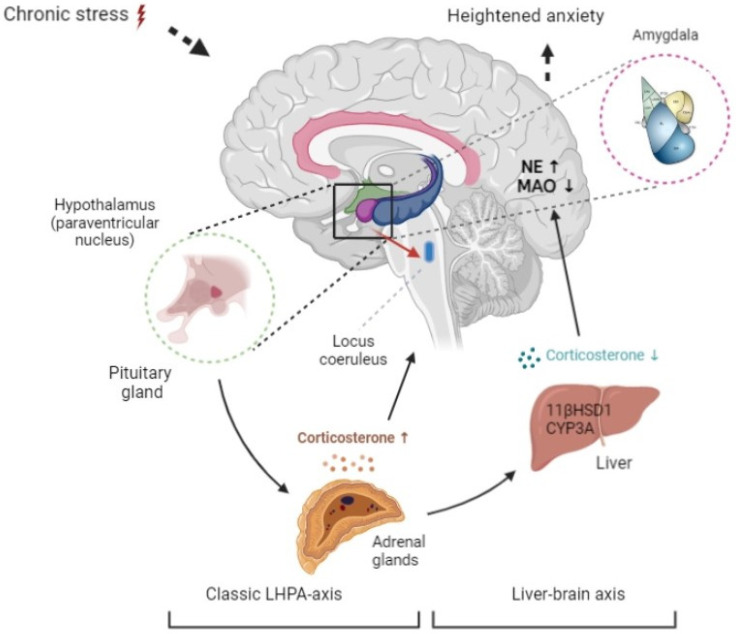
The proposed mechanism for developing anxiety-like behavior in the presence of reduced plasma corticosterone via the liver–brain axis. Initially, activation of the limbic-hypothalamic-pituitary-adrenal system increases corticosterone. This leads to activation of hepatic glucocorticoid-metabolizing enzymes, CYP3A, and 11-βHSD-1, and plasma corticosterone falls. This causes a decrease in brain MAO activity and an increase in brain norepinephrine. As a result, anxiety increases in spite of reduced plasma corticosterone.

**Figure 2 ijms-23-04881-f002:**
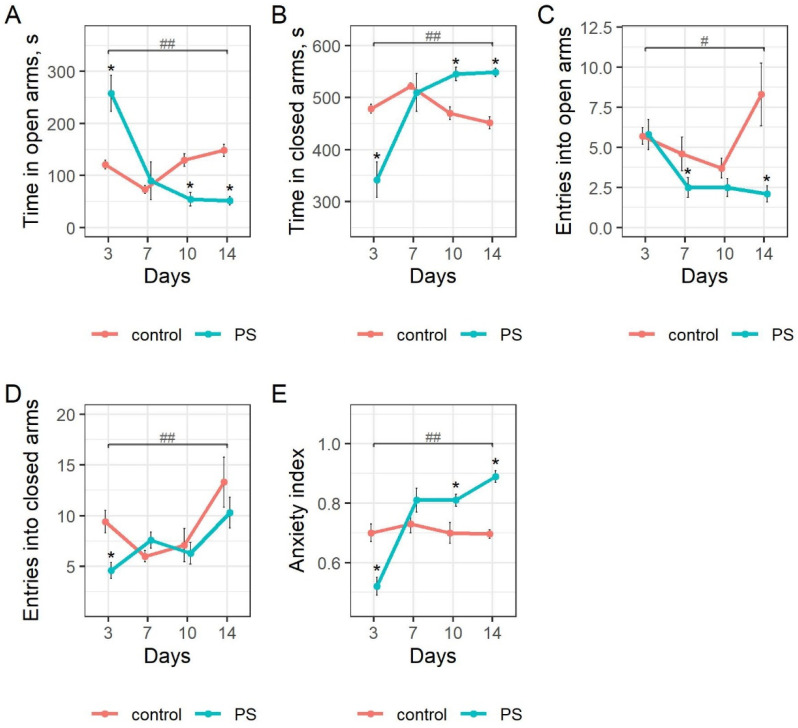
The impact of post-stress time on the behavioral patterns of stressed and control rats in the EPM test. Mann–Whitney comparisons for M ± SD of (**A**) time in open arms, (**B**) time in closed arms, (**C**) entries into open arms, (**D**) entries in closed arms, (**E**) anxiety index. * *p* < 0.05 for differences between means of respective control and PS groups. # *p* < 0.05, ## *p* < 0.01 for differences between means of PS group at post-stress days 3 and 14.

**Figure 3 ijms-23-04881-f003:**
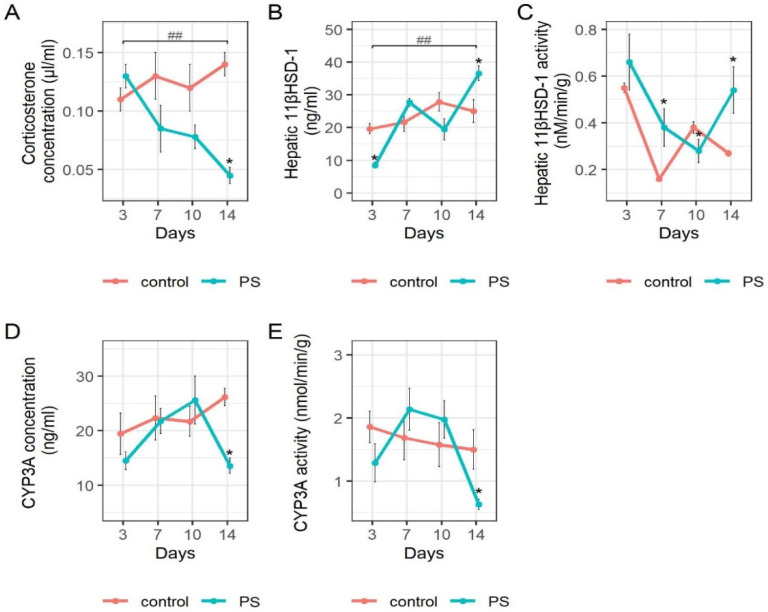
The impact of post-stress time on plasma corticosterone concentrations and on glucocorticoid-metabolizing enzymes protein concentration and activity in the liver. Mann–Whitney comparisons of M ± SD for (**A**) plasma corticosterone concentration, (**B**) hepatic 11-βHSD-1 protein concentration, (**C**) hepatic 11-βHSD-1 activity, (**D**) hepatic CYP3A protein concentration, (**E**) hepatic CYP3A activity. * *p* < 0.05 for differences between means of respective control and PS groups. ## *p* < 0.01 for differences between means of PS group at post-stress days 3 and 14.

**Figure 4 ijms-23-04881-f004:**
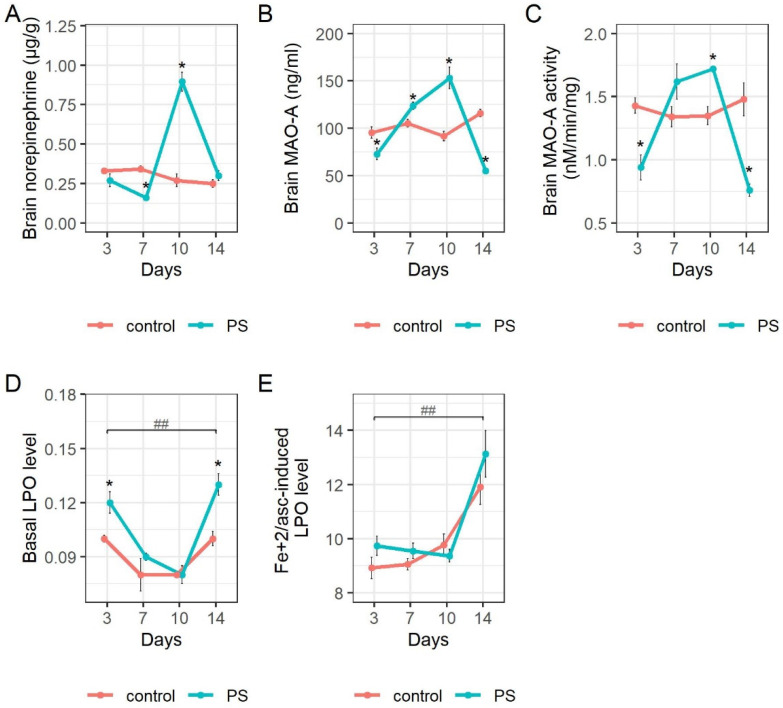
The impact of post-stress time on brain norepinephrine concentration, MAO-A protein concentration and activity, and LPO levels. Mann–Whitney comparisons of M ± SD for (**A**) brain norepinephrine concentration, (**B**) brain MAO-A concentration, (**C**) brain MAO activity, (**D**) basal LPO content, (**E**) Fe^2+^/ascorbate induced LPO content. * *p* < 0.05 for differences between means of respective control and PS groups. ## *p* < 0.01 for differences between means of PS group at post-stress days 3 and 14.

**Figure 5 ijms-23-04881-f005:**
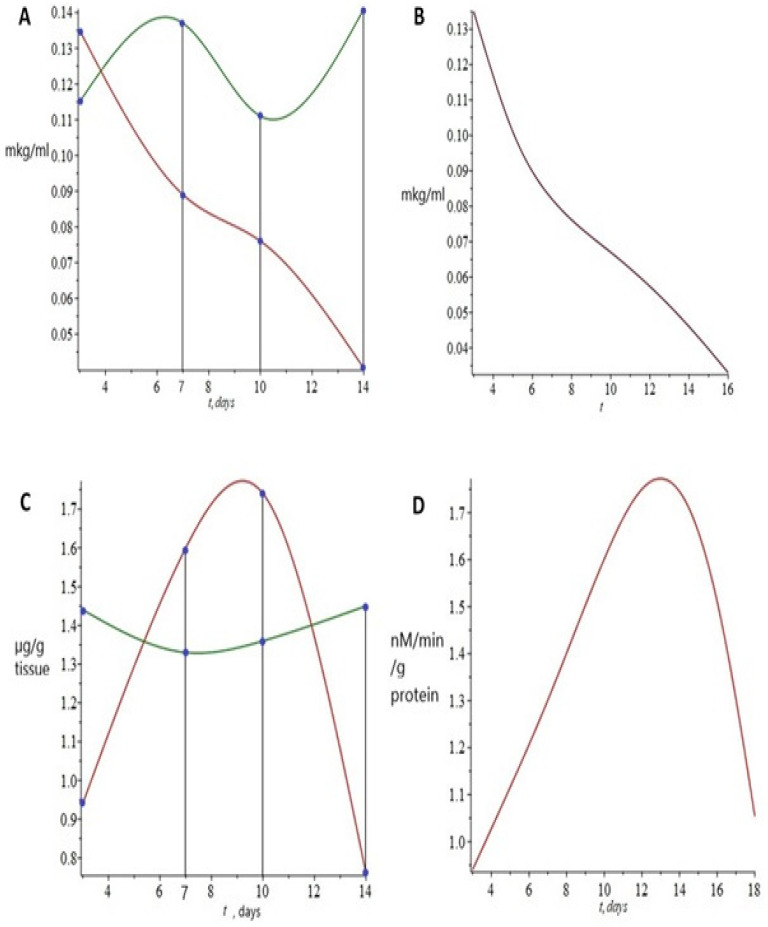
Plasma corticosterone concentration (**A**) and brain MAO-A activity (**C**) during the post-stress period for stressed rats (red points) and control rats (green points). The points have been connected by spline interpolation (red and green lines). Plasma corticosterone concentration (**B**) and brain MAO-A activity (**D**) predicted by the mathematical model for the stressed rats are illustrated by continuous red lines.

**Figure 6 ijms-23-04881-f006:**
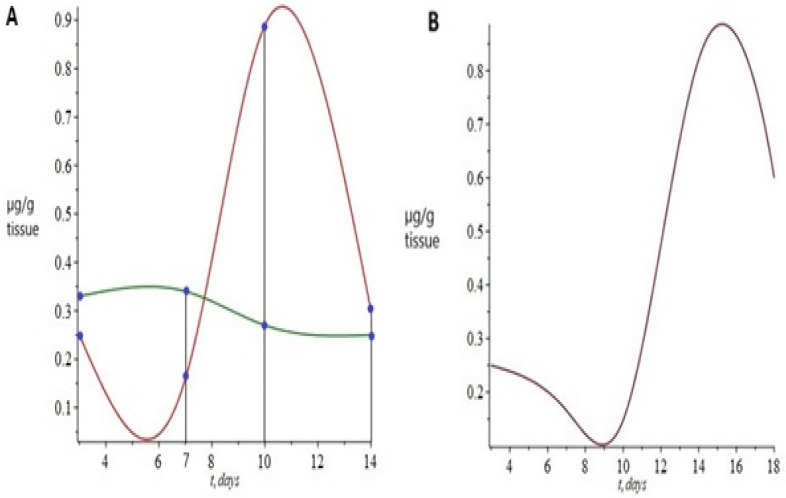
(**A**) Brain norepinephrine concentration of stressed rats (red points) and control rats (green points) during the post-stress period. The points have been connected by spline interpolation (red and green lines). (**B**) Brain norepinephrine concentration predicted by the mathematical model for the stressed rats is illustrated by a continuous red line.

**Figure 7 ijms-23-04881-f007:**
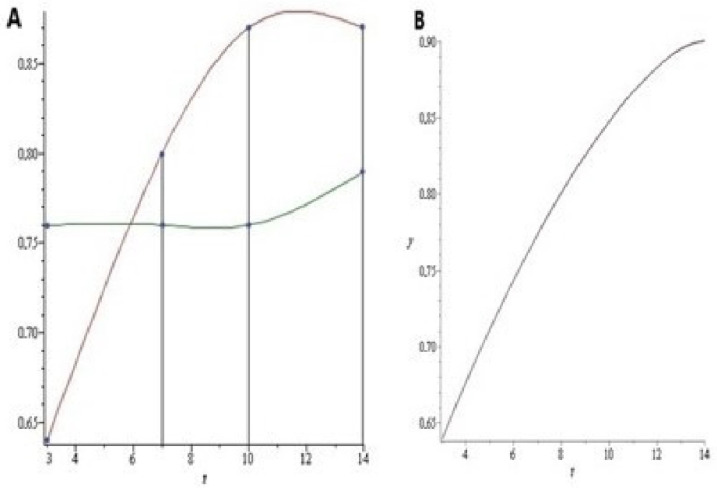
(**A**) Anxiety index of stressed rats (red points) and control rats (green points) during the post-stress period. The points have been connected by spline interpolation (red and green lines). (**B**) Anxiety index predicted by the mathematical model for the stressed rats is illustrated by a continuous red line.

**Figure 8 ijms-23-04881-f008:**
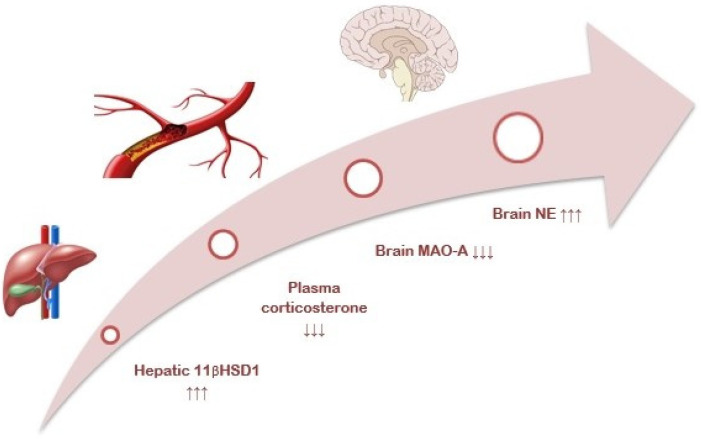
11-βHSD-1-dependent pathway of anxiety-like behavior. In the post-traumatic period, activation of 11-βHSD-1 and reduction in plasma corticosterone occur. In turn, lower plasma corticosterone is associated with a reduction in brain MAO-A activity and with an increase in brain norepinephrine concentration.

**Figure 9 ijms-23-04881-f009:**
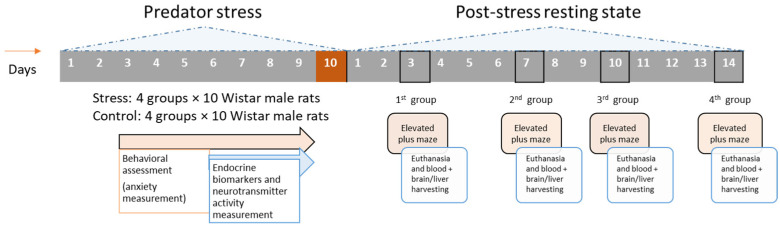
Timeline of experimental procedures. All experimental groups received 10 days of PSS, and all control groups received 10 days of sham PSS stress. Experimental and control groups were tested for behavior and then killed for biochemical assays of blood, brain, and liver at 3, 7, 10, and 14 days after PSS or sham PSS, respectively.

## Data Availability

Data available on request from the authors.

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
