# Peer review of "The Link between Activities of Hepatic 11beta-Hydroxysteroid Dehydrogenase-1 and Monoamine Oxidase-A in the Brain Following Repeated Predator Stress: Focus on Heightened Anxiety"

_ijms, 2022, doi:10.3390/ijms23094881_

Round 1
Reviewer 1 Report
The paper is much improved and worthy of publication
Author Response
Reviewer 1: The paper is much improved and worthy of publication
Response to Reviewer 1
We regret the language errors, and we appreciate you calling those errors to our attention. We have corrected the errors, and we have carefully proofread this revised version.
Reviewer 2 Report
9 April 2021
Review on the manuscript titled “The link between activities of hepatic 11beta-hydroxysteroid dehydrogenase-1 and monoamine oxidase-A in the brain following chronic predator stress: Focus on heightened anxiety” by Lapshin M et al., submitted to International Journal of Molecular Sciences (IJMS)
Manuscript ID: ijms-1678547
Dear Authors,
I am very pleased to see that the Authors have welcomed my suggestions and have clarified most of the questions I raised in my first round of this review. I believe that this original research article does an excellent work developing a mathematical model that can describe the dynamics of the biochemical variables and factor(s) responsible for the development and dynamics of anxiety.
I still suggest increasing the number of references to support the authors’ argumentation. Original research article like this should list at least more than 50 references. Thus, I have few last minor suggestions to do, to further improve the theoretical background of the present article and its argumentation on psychiatric mood disorders (i.e., depression, anxiety and PTSD).
In this regard, I believe that in the ‘Discussion’ section, Authors should have taken the opportunity to move beyond investigating pathogenesis of anxiety disorders including inflammation and discussed theoretical and methodological avenues in need of refinement. In this regard, I suggest to add evidence to suggest a path forward, that focused on neural substrates of anxiety disorder and PTSD, specifically on frontal lobe dysfunction, and on related effects on patients’ memory and learning impairments (https://doi.org/10.3390/brainsci11081023; https://doi.org/10.1007/s43440-020-00067-5https://doi.org/10.3390/biomedicines9050517; https://doi.org/10.3390/biomedicines9070715; https://doi.org/10.3390/biomedicines9080994; https://doi.org/10.3390/biomedicines9101293; doi: 10.3390/biomedicines9020112; https://doi.org/10.1038/s41380-021-01326-4; https://doi.org/10.1038/s41380-021-01326-4).
Moreover, it would be useful to also have a general overview on the application of new techniques, such as the Non-invasive brain stimulation techniques (NIBS), in the treatment of mood disorders. For this reason, I would suggest some crucial references that will methodologically fit with the present manuscript, as evidence for the implementation of these new methods in the treatment of mental disorder in humans (https://doi.org/10.1016/j.neubiorev.2021.04.036; https://doi.org/10.1016/j.jad.2021.02.076; https://doi.org/10.3390/biomedicines10030627).
Overall, this is a timely and needed study, and I look forward to seeing further study on this issue by these authors in the future.
I am always available for other reviews of such interesting and important articles. I look forward to seeing further study on this issue by these authors in the future.
Thank You for your work.
I declare no conflict of interest regarding this manuscript.
Best regards,
Reviewer
Author Response
Reviewer 2
Dear Authors,
I am very pleased to see that the Authors have welcomed my suggestions and have clarified most of the questions I raised in my first round of this review. I believe that this original research article does an excellent work developing a mathematical model that can describe the dynamics of the biochemical variables and factor(s) responsible for the development and dynamics of anxiety.
I still suggest increasing the number of references to support the authors’ argumentation. Original research article like this should list at least more than 50 references. Thus, I have few last minor suggestions to do, to further improve the theoretical background of the present article and its argumentation on psychiatric mood disorders (i.e., depression, anxiety and PTSD).
In this regard, I believe that in the ‘Discussion’ section, Authors should have taken the opportunity to move beyond investigating pathogenesis of anxiety disorders including inflammation and discussed theoretical and methodological avenues in need of refinement. In this regard, I suggest to add evidence to suggest a path forward, that focused on neural substrates of anxiety disorder and PTSD, specifically on frontal lobe dysfunction, and on related effects on patients’ memory and learning impairments (https://doi.org/10.3390/brainsci11081023; https://doi.org/10.1007/s43440-020-00067-5https://doi.org/10.3390/biomedicines9050517; https://doi.org/10.3390/biomedicines9070715; https://doi.org/10.3390/biomedicines9080994; https://doi.org/10.3390/biomedicines9101293; doi: 10.3390/biomedicines9020112; https://doi.org/10.1038/s41380-021-01326-4; https://doi.org/10.1038/s41380-021-01326-4).
Moreover, it would be useful to also have a general overview on the application of new techniques, such as the Non-invasive brain stimulation techniques (NIBS), in the treatment of mood disorders. For this reason, I would suggest some crucial references that will methodologically fit with the present manuscript, as evidence for the implementation of these new methods in the treatment of mental disorder in humans (https://doi.org/10.1016/j.neubiorev.2021.04.036; https://doi.org/10.1016/j.jad.2021.02.076; https://doi.org/10.3390/biomedicines10030627).
Overall, this is a timely and needed study, and I look forward to seeing further study on this issue by these authors in the future.
I am always available for other reviews of such interesting and important articles. I look forward to seeing further study on this issue by these authors in the future.
Thank You for your work.
I declare no conflict of interest regarding this manuscript.
Best regards,
Response to Reviewer 2
We greatly appreciate your high appraisal of our manuscript.
Thank you very much for your valuable comments and recommendations. We applied them in editing the manuscript. We completely agree with your advice to expand the Discussion by including comments on the pathogenesis of anxiety disorders and neuroinflammation. Certainly, you are absolutely right that it would be relevant to note new approaches to understanding PTSD. We have added this to the latter part of the Discussion. Special thanks for relevant and very interesting references. We enjoyed making ourselves familiar with these studies and cited them.
Reviewer 3 Report
- "Grafic abstract" - misspelled.
- Please confirm Figure 1 is fine concerning copyright issues.
- "The oder of result presentation" - spelling error.
- "Therefore we firstly indicated the anxiety-like behavior and after we described data concerning the liver and finished by brain data" - unsure what this means. Language still in need of close reading and extensive edits.
- "repeated stress" and "chronic stress" are not the same, please do not use them interchangeably.
Author Response
Response to Reviewer 3
Thank you very much for your positive review and for finding our shortcomings. We believe we have corrected them.
- Grafic abstract" - misspelled. The misspelling error has been corrected.
- Please confirm Figure 1 is fine concerning copyright issues.
We confirm that this figure has never been published before, and publishing it does not require special permission.
- The oder of result presentation" - spelling error.
We have removed this sentence.
- Therefore we firstly indicated the anxiety-like behavior and after we described data concerning the liver and finished by brain data" - unsure what this means. Language still in need of close reading and extensive edits.
We have removed this unclear sentence as it was redundant. We regret the language errors in the previous version, and we have carefully proofread the text of this revised version.
repeated stress" and "chronic stress" are not the same, please do not use them interchangeably.
We agree that “repeated stress” and “chronic stress” must not be used interchangeably. In our experimental protocol, rats were exposed to repeated stress, so wherever appropriate we have changed “chronic stress” to “repeated stress.”
Round 2
Reviewer 3 Report
Thank you for the revisions.
This manuscript is a resubmission of an earlier submission. The following is a list of the peer review reports and author responses from that submission.
Round 1
Reviewer 1 Report
Lapshin and colleagues investigated the link between activities of 11beta-hydroxysteroid dehydrogenase-1 in the liver, and MAO-A in the brain in Chronic Post- 3 traumatic Stress paradigm. Despite the results might be interesting, this work needs to be substantially improved.
- The Authors speculate a lot talking about PTSD. However, the experimental model they used is questionable because mostly at the behavioral level, PTSD cannot be restricted to only anxiety (They only performed an EPM test). I really suggest to read the DSM-5 to understand what is PTSD.
- I also strongly suggest to take into consideration a better experimental model to mimic PTSD (traumatic stress susceptibility and resilience). Indeed, it is crucial to remark that stress-related disorders develop only in vulnerable individuals. Moreover, resilience is a dynamic process that need to be deeply studied to unravel new information.
- The material and method are poorly described. It is difficult to understand the results.
Reviewer 2 Report
The authors present a mathematical model to study the intensity of glucocorticoid tissue metabolism, brain MAO-A activity, norepinephrine concentrations, and the concentration of corticosterone in the blood. It is a rather difficult-to-read study and the findings were not well-articulated.
Specific comments:
- The abstract is too long. As per journal guidelines, it should be a total of about 200 words maximum.
- Please change "To deter-mine MAO-A" to "To determine MAO-A".
- Were the mice fed a controlled diet throughout the entire study duration? Please specify.
- As this study involved the use of vertebrates, it is important to mention exactly how the animals were handled. In order to prevent undue suffering, ethical considerations in animal studies are important. It is important for the authors to state any guidelines followed in the conduct of this study. This should be mentioned under the 'methods' section.
- ".... Ethical Committee for Animal Experiments of South Ural State University, Chelyabinsk, Russia" - please provide the actual IRB approval/study number.
- What does "p=0.019U" mean?
- I am not sure that the findings actually support the presence of a molecular pathway from glucocorticoid-metabolizing enzymes to monoamine-A metabolizing enzymes in PTSD development. The findings from the animal model are unable to establish causation and the elevated plus maze (EPM) test only assesses anxiety-related behavior in rodent models, it says nothing about PTSD development or pathogenesis.
- In fact, while EPM is a commonly employed animal behavioral model of anxiety, there are several issues concerning the validity of the model. Selective serotonin reuptake inhibitors and tricyclic antidepressants, which are clinically indicated for the treatment of anxiety disorders, do not produce a stable anxiolytic effect on EPM testing.
- From an epigenetic perspective, early life stress may also result in durable changes in the glucocorticoid receptor gene, giving rise to a pro-inflammatory phenotype. This was not examined.
- "corti-costerone" has no hyphen.
- In recent years, the combinations of gas chromatography-quadrupole time of flight mass spectrometry (GC-Q-TOF/MS) and liquid chromatography-quadrupole time of flight mass spectrometry (LC-Q-TOF/MS) have been applied successfully in numerous metabolomics studies to achieve more sensitive and accurate metabolic profiling and conduct of mechanistic investigations (citation: pubmed.ncbi.nlm.nih.gov/30056340). The use of GC-Q-TOF/MS is a possible area of future work the authors should mention in the discussion section to better identify the metabolite changes and pathways involved.
- The underlying data should be made publicly available. When specific legal or ethical requirements prohibit public sharing of a dataset, authors must indicate how researchers may obtain access to the data.
Reviewer 3 Report
Review on the manuscript titled “The link between activities of 11beta-hydroxysteroid dehydro-genase-1 in the liver, and MAO-A in the brain in Chronic Post-traumatic Stress paradigm” by Lapshin M et al., submitted to International Journal of Molecular Sciences (IJMS)
Manuscript ID: ijms-1515071
Post-traumatic stress disorder (PTSD) is characterized by low circulating cortisol, high brain norepinephrine, different activity levels of hepatic 11-β-hydroxysteroid dehydrogenases and monoamine oxidase A. The authors developed a mathematical model previously. The authors measured parameters in social defeat models of rats to evaluate the validity of the model. The results showed the mathematical model is in line with an animal model of PTSD.
Please consider the following parts:
- A graphical abstract is highly recommended.
- Page 1, Abstract: Please clearly present background (known, unknown), the purpose of this study, methods used in this study, results, and conclusion proportionally with 200-220 words.
- Page 1, Keywords: Please present up to ten keywords.
- Pages 2,3, Introduction:
- Please rearrange the introduction describing PSTD (definition, epidemiology, pathology), translational animal model, current understanding in pathogenesis including the liver-brain axis, mathematical model, and the purpose of this study. Suggested reference: https://doi.org/10.3390/biomedicines9101293.
- Short descriptions of inflammation and probably mitochondrial function in PTSD may enrich this manuscript. Suggested reference: https://doi.org/10.3390/biomedicines9070734.
- A figure summarizing the introduction is recommended.
- Pages 3-10, Results:
- Please present the results in a logical order. Probably the sequence can be described in introduction.
- Please present statistical values including p value in tables.
- Pages 9-12, Discussion: Please discuss the previous and present study, the limitation and weaknesses in current study, potentials, the ultimate goal, research or knowledge needed to achieve, the biggest challenge in this goal, and future research directions, among others.
- Pages 12-14, Materials and Methods: The description of behavioral testing is missing. Please present the methods in order presented in results.
The manuscript contains six figures, no table, and 87 references. The manuscript carries important value presenting the validity of a mathematical mode in animal model of PTSD.

Reviewer 4 Report
In the present study, the Authors proposed the presence of a molecular pathway from glucocorticoid-metabolizing enzymes to monoamine-A metabolizing enzymes in PTSD development. Therefore, They evaluated the liver 11beta HSD1- and brain MAO-A activities with added measurements of blood corticosterone and brain noradrenaline concentrations in an experimental PTSD paradigm. They obtained the experimental curves relating plasma corticosterone and noradrenaline concentrations to brain MAO-A activity.
Overall, I found the present study timely, very interesting and scientifically sound. I have only some minor suggestions aimed to improve the high quality of the paper and these are outlined below:
1) In the Introduction, it's worthy to explain what is post-traumatic stress disorder (PTSD) with appropriate references (see dois: 10.9758/cpn.2021.19.4.780 and 10.1080/13651501.2019.1699575).
2) In the process of PTSD chronification might be also involved the glutamatergic system? Please add a brief consideration on this point.